# A Multidisciplinary Approach to the Classification and Management of Intestinal Failure: Knowledge in Progress

**DOI:** 10.3390/diagnostics14192114

**Published:** 2024-09-24

**Authors:** Sol Ramírez-Ochoa, Luis Asdrúval Zepeda-Gutiérrez, Mauricio Alfredo Ambriz-Alarcón, Berenice Vicente-Hernández, Gabino Cervantes-Guevara, Karla D. Castro Campos, Karla Valencia-López, Gabino Cervantes-Pérez, Mariana Ruiz-León, Francisco Javier Hernández-Mora, Tania Elizabeth Cervantes-Nápoles, María Elena Flores-Villavicencio, Sandra O. Sánchez-Sánchez, Enrique Cervantes-Pérez

**Affiliations:** 1Department of Internal Medicine, Hospital Civil de Guadalajara Fray Antonio Alcalde, Centro Universitario de Ciencias de la Salud, Universidad de Guadalajara, Guadalajara 44280, Jalisco, Mexico; sramirez@hcg.gob.mx (S.R.-O.); luis-zent@hotmail.com (L.A.Z.-G.); dra.berenicevicente@gmail.com (B.V.-H.); kdcc1124@gmail.com (K.D.C.C.); karlavalencialopez64@gmail.com (K.V.-L.); gabino-1994@hotmail.com (G.C.-P.); ecervant@its.jnj.com (M.R.-L.); sandyolivia.ss@gmail.com (S.O.S.-S.); 2División de Servicios Intermedios, Hospital Civil de Guadalajara Fray Antonio Alcalde, Guadalajara 44280, Jalisco, Mexico; mau_ambriz@hotmail.com; 3Department of Gastroenterology, Hospital Civil de Guadalajara Fray Antonio Alcalde, Guadalajara 44280, Jalisco, Mexico; gabino_guevara@hotmail.com; 4Department of Welfare and Sustainable Development, Centro Universitario del Norte, Universidad de Guadalajara, Colotlán 46200, Jalisco, Mexico; 5Department of Human Reproduction, Growth and Child Development, Health Sciences University Center, Universidad de Guadalajara, Guadalajara 44280, Jalisco, Mexico; frank.gine@gmail.com; 6Department of Philosophical, Methodological and Instrumental Disciplines, Centro Universitario de Ciencias de la Salud, Universidad de Guadalajara, Guadalajara 44340, Jalisco, Mexico; tania.cervantes@academico.udg.mx; 7Departament of Social Sciences, Centro Universitario de Ciencias de la Salud, Universidad de Guadalajara, Guadalajara 44340, Jalisco, Mexico; marlencilla27@hotmail.com; 8Centro Universitario de Tlajomulco, Universidad de Guadalajara, Tlajomulco de Zúñiga 45641, Jalisco, Mexico

**Keywords:** intestinal failure, short bowel syndrome, parenteral nutrition, GLP-2 analogs, interdisciplinary care

## Abstract

Intestinal failure (IF) is a debilitating condition characterized by the insufficient function of the gastrointestinal tract to absorb nutrients and fluids essential for life. This review consolidates recent advancements and challenges in managing IF among adult and pediatric populations, highlighting differences in etiology, management, and outcomes. Over the recent years, significant strides have been made in the nutritional and medical management of IF, significantly reducing mortality rates and improving the quality of life for patients. Key advancements include the development and availability of glucagon-like peptide-2 (GLP-2) analogs, improved formulations of parenteral nutrition, and the establishment of specialized interdisciplinary centers. Short bowel syndrome (SBS) remains the predominant cause of IF globally. The pediatric segment is increasingly surviving into adulthood, presenting unique long-term management challenges that differ from adult-onset IF. These include the need for tailored nutritional support, management of IF-associated liver disease, and addressing growth and neurodevelopmental outcomes. The therapeutic landscape for IF continues to evolve with the development of new treatment modalities and better understanding of the condition’s pathophysiology. However, disparities in treatment outcomes between children and adults suggest the need for age-specific management strategies. This review underscores the importance of a nuanced approach to IF, incorporating advancements in medical science with a deep understanding of the distinct needs.

## 1. Introduction

Intestinal failure (IF) is characterized by a decrease in gastrointestinal function below the level required for proper digestion and absorption of nutrients, fluids, and development needs. This often leads to a protracted dependence on specialized nutrition support [1]. Patients with IF experience functional and structural compensatory mechanisms in the gut, which are designed to enhance the absorptive capacity of the intestine. Effective management of this condition necessitates an interdisciplinary approach that focuses on promoting intestinal rehabilitation while also addressing the consequences of IF. These consequences include chronic diarrhea, malabsorption, loss of appetite, excessive growth of bacteria in the small intestine, intolerance to enteral feeding, and complications associated with long-term parenteral nutrition (PN) [2]. Several recent studies have demonstrated progress in parenteral and enteral nutritional therapy, as well as enhanced comprehension of the difficulties linked to long-term PN and IF. The European Society for Clinical Nutrition and Metabolism (ESPEN) has just emitted an updated practical guideline that offers a clinical basis for managing this challenging condition [3]. This review will discuss recent discoveries on the prevalence and distribution of IF, advancements in PN, the significance of metabolic bone disease (MBD) and small bowel bacterial overgrowth (SBBO), and the progress made in the utilization and comprehension of glucagon-like protein 2 (GLP-2) analog drugs.

## 2. Defining IF

ESPEN provided criteria and classifications for adults with IF in 2014, using the Delphi method. Their definition necessitated the existence of two criteria: “reduced assimilation of macronutrients and/or water and electrolytes as a result of impaired gastrointestinal function” and the “requirement for intravenous supplementation”. This definition excludes patients who have underlying conditions that are not related to the condition being classified as IF. Examples of such unrelated conditions include reduced food intake but normal gut function, altered gut function but adequate intestinal absorption, impaired intestinal absorption but still being maintained on enteral nutrition, refusal of enteral nutrition, and the need for only vitamin or trace element supplementation. (Table 1).

Regarding the pediatric population, the North American Society for Pediatric Gastroenterology, Hepatology and Nutrition (NASPGHAN) developed international guidelines in 2017 that provide detailed strategies for the management of pediatric IF and SBS. NASPGHAN defined IF as the need for PN for more than 60 days due to intestinal disease, dysfunction, or resection. Individuals who have less than 25% of the predicted length of their colon after bowel resection are considered to meet the criteria for IF immediately, without having to wait for 60 days of PN dependence. In addition, they advised that patients should be referred to a multidisciplinary intestinal rehabilitation program (IRP) to improve both short- and long-term outcomes. In 2021, the American Society for Parenteral and Enteral Nutrition (ASPEN) published recommendations that provide a defined terminology for pediatric intestinal failure (IF). IF is defined as the need for PN support for more than 60 days within the last 74 days, specifically related to intestinal injury or resection.

## 3. Epidemiology and Classification

### 3.1. Epidemiology in Adult Population

SBS is the predominant etiology of IF on a global scale. Among people with intestinal failure (IF) enrolled in a multicenter worldwide registry undertaken by ESPEN, SBS accounted for 67% of cases of IF. This register consisted of 65 locations across 22 nations, with a total of 1880 patients. Three distinct kinds of SBS were identified: Type 1, SBS-J, characterized by a short bowel with an end jejunostomy; Type 2, SBS-JC, characterized by a short bowel with a jejuno-colic anastomosis; and Type 3, SBS-JIC, characterized by a short bowel with a jejuno-ileal anastomosis and preservation of the ileocecal valve. An overall female-to-male ratio of 2:1 was observed across all categories. The age distribution was comparable across different forms of SBS, with 68.2% of Type 1, 72% of Type 2, and 62.8% of Type 3 cases observed in adults aged 50 or older [4].

The primary etiologies of SBS vary, with Crohn’s disease (CD) accounting for the majority of instances leading to SBS-J, whereas mesenteric ischemia is the predominant cause of SBS-JC and SBS-JIC. Consequently, CD and MI are the prevailing factors responsible for SBS in adults globally. The nations with the highest prevalence of SBS-J were unsurprisingly those where CD was the predominant underlying condition. This includes 82.8% of cases in Denmark, 75.3% of cases in the United Kingdom, and 59.7% of cases in the United States. France and Italy had higher rates of mesenteric ischemia and correspondingly higher rates of SBS-JC. These variations among nations likely indicate distinct approaches to managing CD and MI on a global scale. Although every patient in the research necessitated intravenous support, there were significant variations in its utilization. In patients with SBS-JC, intravenous support (IVS) was consistently required for a duration exceeding 3 years in 61.4% of cases and SBS-JIC 52.3% of the patients. Conversely, less than half of patients with SBS-J required IVS for a duration longer than 3 years. Approximately half of the total cohort necessitated daily intravenous solutions [4].

A countrywide inpatient database in the United States, covering the period from 2005 to 2014, was used to evaluate hospitalization patterns, disease burden, death rates, and healthcare usage. This study demonstrated an elevated prevalence of females (68%) and a significant proportion of patients over the age of 50 (67.8%). The annual hospitalization rate experienced an upward trend during the 10-year research period, with the underlying causes remaining uncertain. Fluid and electrolyte imbalances, particularly hypokalemia and hyponatremia, were the primary factors leading to hospitalizations, accounting for 52.5% of cases. The primary comorbidity observed was infection, specifically sepsis and/or bacteremia, which affected 41.1% of the patients. During the 10-year study period, the average rate of death from any cause while in the hospital was 3.8%. However, there was a noticeable decrease in yearly mortality between 2005 and 2014, which was statistically significant (*p* = 0.03). The two most important characteristics that predict the likelihood of dying in the hospital are sepsis (adjusted odds ratio [aOR], 3.38; 95% confidence interval [CI], 3.02–3.78, *p* < 0.001) and being over the age of 65 (aOR 3.49; 95% CI, 2.68–4.56, *p* < 0.001). Additional autonomous risk factors influencing death during hospitalization encompassed acute malnutrition, congestive heart failure, and simultaneous liver disease [5].

### 3.2. Epidemiology in Pediatric Population

A multicenter, international retrospective study was conducted from 2010 to 2015, involving 443 patients from six pediatric programs, with a focus on pediatric healthcare. The predominant etiologies of intestinal failure were short bowel syndrome (84.9%), intestinal enteropathies (7.9%), and intestinal dysmotility (7.2%). In contrast to the findings in adult studies, a significant majority (61.2%) of the patients were male. Out of the total number of participants (*n* = 213), 48% (*n* = 213) achieved enteral autonomy at the conclusion of the study period. 12% (*n* = 53) underwent transplantation, 9% (*n* = 40) died while receiving parenteral nutrition, and 31% (*n* = 137) continued to depend on parenteral nourishment [6].

Although the mortality rate in pediatric IF is decreasing, comparable to adult studies, the risk variables linked to death and the necessity for transplantation vary. Factors linked to reduced mortality and lower need for transplantation included early diagnosis before the age of 1, the presence of an ileocecal valve, a colon connected to the small bowel, and modest indications of intestinal failure associated liver disease (IFALD) [6,7]. Although mortality and transplantation rates have consistently declined, there has been no corresponding increase in the number of children attaining enteral autonomy. Instead, a greater percentage of youngsters continue to rely on parenteral care for extended durations. These data indicate that a subset of children with SBS may soon become adults who rely on parenteral support. This group will be distinct from the current adult population with SBS, presenting new issues in terms of diagnosis, management, and treatment. The illness mechanisms in this cohort will differ from those observed in current adult individuals with IF. Furthermore, it is crucial to prioritize age-appropriate neurodevelopment, quality of life, pubertal development, and transitions of care [6,7].

### 3.3. Classification

These studies demonstrate significant distinctions between childhood and adult SBS. As surgical, medicinal, and nutritional therapies advance and interdisciplinary intestinal rehabilitation facilities proliferate, the death rates for IF-SBS are progressively increasing. Consequently, the healthcare use for IF-SBS is expected to increase. These studies demonstrate variations in treatment methods, results, and disease attributes on a worldwide level and establish the groundwork for gaining a deeper comprehension of the evolving community of individuals with SBS from childhood to adulthood. A summary of the etiologies and classification of SBS in IF can be found in Table 2.

## 4. Rehabilitation

Rehabilitation in the context of IF is defined as the restoration of intestinal function. The rehabilitation process is considered to begin immediately after bowel resection and is mediated by several factors such as age, blood flow, dietary factors, factors related to intestinal physiology, among others. In addition to these factors, there is an increase in intestinal area (mediated by an increase in the length of the villi and the depth of the crypts), which can increase intestinal absorption. It is estimated that this process takes 1 to 2 years. In general, in adults it is necessary to preserve more than 100 cm of healthy residual small intestine to achieve optimal adaptation. However, adequate adaptation can be achieved with less residual small bowel in the presence of the colon, because the colonic microbiota can ferment carbohydrates to short-chain fatty acids for use as an energy source [8].

## 5. Medical Nutrition Therapy in Patients with IF

### 5.1. Oral Feeding

The dietary therapy should prioritize the management of compensatory hyperphagia. Minimal quantities of luminal nutrients promote intestinal adaption and provide protection against liver disorders and other problems [9]. Effective dietary counseling should be tailored to the individual patient’s preferences in order to maximize adherence. Modifications can be implemented according to the tolerance, symptoms, stool production, and weight. To compensate for malabsorption, it is necessary to increase food intake by a minimum of 50% from the predicted needs. This increase should be divided across 5–6 meals spread throughout the day [10]. It is advisable to consume foods that have a high energy density and a high salt content. Patients should consume salt generously and limit their use of fluids by mouth when eating. Patients with SBS with an intact colon can benefit from a diet that is high in carbohydrates (60%) and low in fat (20%), while also excluding foods high in oxalate (such as peanuts and baked beans). This type of diet has been shown to decrease the loss of calories in the feces, increase the absorption of energy, and minimize the absorption of magnesium, calcium, and oxalate. It is recommended to consume a diet that is rich in medium-chain triglycerides and low in fat, as this helps to decrease the occurrence of steatorrhea and limits the fermentation of carbohydrates [11,12]. The fat/carbohydrate ratio is not a significant factor in patients with end jejunostomy [13,14].

### 5.2. Enteral Nutrition

Enteral nutrition (EN) should be addressed, particularly in individuals with low PN reliance, who are anticipated to be gradually withdrawn from PN. Even in patients with a restricted capacity for total PN discontinuation, EN can provide significant advantages [15]. Performing a percutaneous gastrostomy can be technically problematic due to the changed anatomy and intra-abdominal adhesions that are common in SBS. It is essential to have a thorough discussion about both the potential dangers and advantages, and it is recommended to conduct a trial using a nasogastric tube. Polymeric mixtures are more desirable than elemental formulas due to their lower cost, less hyperosmotic effects, and better tolerance. Nevertheless, research indicates that both formulae exhibit comparable levels of nutritional absorption and fluid/electrolyte loss. Continuous infusion appears to improve the advantages and tolerance of EN. Feeding during the night enhances the overall well-being and allows for the performance of regular daily tasks [16].

### 5.3. Parenteral Nutrition

After undergoing surgical removal, the majority of patients need PN to facilitate healing and provide the proper balance of fluids and electrolytes. A significant number of these patients necessitate extended periods of PN and/or intravenous fluids. Efforts should be focused on optimizing the PN formula to aid in the transition to EN by tube feeding and/or oral diet [17]. It is recommended that patients undergo PN cycling for a duration of 12 to 16 h. This allows them to be free from being connected to an intravenous and promotes the ingestion of food orally. In instances with severe SBS, it may be necessary to administer PN and/or intravenous fluids for extended durations in order to avert dehydration and safeguard the kidneys from harm. The utilization of PN and intravenous fluids does not diminish the significance of commencing nourishment through the gastrointestinal tract. The remaining portion of the intestine undergoes adaptation following surgical removal, in which it adjusts to maintain nutritional balance through physiological, cellular, and molecular processes [18,19]. Introducing complex luminal nutrients early on is crucial for stimulating gut adaption [20] (Figure 1).

Although, as noted above, most patients who require surgical removal of a portion of the bowel will require temporary PN, the goal is to promote oral intake, and therefore only those patients who cannot maintain adequate nutritional status through the gastrointestinal tract should be considered for home PN. It is crucial for the patient to comprehend the nature of this procedure and to demonstrate adherence to the therapy. Thorough discharge preparation is essential to guarantee a secure and smooth transfer. The patient should have suitable vascular access, such as a tunneled catheter, port, or peripherally placed central catheter line. The prescription of the PN formula should aim to provide the patient with sufficient nutrition without causing overfeeding, and it should be cycled whenever feasible. Home PN carries the risk of many problems. Therefore, it is crucial to prioritize the prevention of problems associated with PN-dependency. To reduce complications, it is advisable to maximize oral nutrition, avoid and eradicate sepsis, educate the patient on proper line maintenance, and utilize antibiotic and ethanol locks when necessary. It is important to identify and treat any coagulation problems in order to prevent the formation of blood clots. Regular monitoring of these patients is necessary to ensure the therapy is safe and effective, and to determine if they are ready to start weaning [3].

### 5.4. Metabolic Bone Disease

Another potential consequence of IF is metabolic bone disease (MBD), which encompasses a range of conditions resulting from disruption to bone homeostasis. Multifactorial pathogenesis is a hallmark of MBD in patients with IF. Among the key triggering factors are malabsorption of calcium, phosphorus, and magnesium; vitamin D deficiency; and inflammatory diseases underlying gastrointestinal conditions. It is of vital importance to monitor for signs and symptoms of MBD in both adult and pediatric patients with IF, with particular attention to those who are on long-term parenteral nutrition [21].

It is of the utmost importance to monitor bone health in patients with long-term home parenteral nutrition, as a combination of factors, including vitamin D deficiency, underlying diseases, and inadequate calcium and phosphate management, can lead to the development of metabolic bone disorders. In fact, up to 46% of these patients will develop osteoporosis [22].

## 6. Medical Management in IF

The introduction of teduglutide, a GLP-2 hormone analog, significantly transformed the treatment of IF caused by SBS by enhancing the ability of patients with SBS to tolerate enteral feeding [13,23]. To assess the potential long-term effects of teduglutide following discontinuation of treatment, Zaczek et al. outlined clinical outcomes for patients who had prior teduglutide usage. This retrospective, multicenter study comprised thirteen patients with SBS who had been treated with teduglutide between 2009 and 2013 and continued to be on HPN for nine years following the completion of treatment. Variations in parenteral support were quantified in terms of weekly PN volume. Parenteral support did not differ substantially during the initial four years following the discontinuation of teduglutide. The average volume of intravenous fluids required by all patients increased significantly (*p* = 0.036) by the 5-year mark. At five years, there was no statistically significant difference between the prescribed PN volume and the PN need prior to treatment. 92.3% (*n* = 12) of the cohort experienced a substantial increase in PN volume at the 9-year mark, with values ranging from 26.8% to 169.2% (median: 76.1%, *p* < 0.001) relative to the volume requirement at the conclusion of teduglutide treatment. Additionally, 54% (*n* = 7) of patients required a median increase in PN volume of 21.2% relative to their pre-treatment volume. The volume of the remaining six patients necessitated a median reduction of 26% in comparison to their pre-treatment baseline [14]. These results were consistent with those of earlier studies that reported comparable variations in parenteral support subsequent to the cessation of treatment. It is crucial to develop additional metrics that can predict which patient cohorts will maintain enteral autonomy or experience a persistent decline in parenteral support following drug cessation, as well as whether the drug can be discontinued entirely.

In 2012, the FDA approved teduglutide for the treatment of adults with SBS who are dependent on parenteral nutrition despite adequate medical therapy. Although its use in the pediatric population with SBS is not within the FDA-approved label indications, a 12-week clinical study in the pediatric population demonstrated a reduction in PN volumes and four patients achieving complete independence from PN [22,24].

Furthermore, ongoing trials are being conducted to determine the safety and effectiveness of Apraglutide, a new GLP-2 analog that has a longer duration of action compared to teduglutide. Although the initial data shows promise, the long-term results are yet uncertain. The convenience and adherence implications of a weekly dosage regimen, as opposed to a daily one, could have a substantial impact on the quality of life for this group of patients [25].

Intestinal rehabilitation is essential for the medical treatment of SBBO and persistent diarrhea, together with the use of GLP2 analogues. Patients with intestinal frequently experience structural, mucosal, and motility abnormalities that make them more susceptible to bowel dilatation and small bowel bacterial overgrowth. Culbreath et al. performed a retrospective analysis at a single medical center to evaluate the ability of young patients with intestinal failure and short bowel syndrome with small bowel bacterial overgrowth to tolerate enteral feeding, their growth, and their usage of antibiotics. A total of fifty-six individuals getting therapy for SBBO were identified and underwent endoscopic assessment, during which duodenal aspirates were collected. SBBO was defined as a concentration of greater than 10^5 colony-forming units per milliliter (CFU/mL). In 46% of instances (*n* = 26), antibiotics remained unchanged following endoscopy, while in 54% (*n* = 30) medicines were modified. Furthermore, there was a substantial decrease in emesis/feeding intolerance (from 58.9% to 23.2%, *p* < 0.001) and GI bleeding (from 19.6% to 3.6%, *p* < 0.01) in both subgroups. Additionally, there was a noticeable enhancement in stool output in the subgroup where antibiotics had been modified (46.2% before the scope compared to 15.4% after the scope, *p* = 0.005). There was a significant improvement in BMI-for-age z scores in the entire group, with a change from 0.03 to 0.27 (*p* = 0.03). Weight-for-age z scores also showed a trend towards statistical significance, changing from −0.86 to −0.65 (*p* = 0.05). Although this study has limitations due to its retrospective nature and short follow-up period, it is highly probable that treating SBBO leads to enhanced enteral tolerance, reduced gastrointestinal symptoms, and improved weight gain and growth. Further research is required to fully comprehend the impact of small bowel bacterial overgrowth (SBBO) treatment on PN, as well as the significance of antibiotic regimens guided by culture [26].

## 7. Surgical Challenges in Patients with IF

Preserving intestinal length is crucial for individuals who undergo extensive bowel resection to increase the likelihood of achieving independence from PN. A second laparotomy performed 24–48 h after the initial emergency procedure provides an opportunity to reassess the condition of the bowel that may have uncertain viability. This helps evaluate if any more length of the bowel can be saved [27]. Transluminal stents have been used in children to preserve intestinal length in cases with patchy necrotizing enterocolitis (NEC) or type IV intestinal atresia [28]. This surgery entails the creation of a proximal jejunostomy to redirect fecal contents, followed by the insertion of a small-caliber transluminal stent through all the distal segments. The various segments are joined together using two or three sutures, but no formal connection is made to reduce the risk of narrowing. Once the bowel that has been treated with a stent has had enough time to mend and regain its strength (about 6 weeks), patency is assessed with contrast fluoroscopy, and a second surgery is completed to place the bowel back into continuity [28]. An advantage of this approach is that it can increase the length of the residual bowel as the segments continue to grow, taking use of the natural growth potential in newborns. The utilization of transluminal stents has not been documented in the literature pertaining to adults.

Patients who have reached a plateau in their ability to wean off parenteral support or who are experiencing significant complications such as SBBO resistant to medical treatment, recurrent sepsis, or IFALD, are candidates for autologous bowel reconstruction procedures. The three primary types of autologous gut reconstruction treatments are: intestinal lengthening techniques; delayed transit; and restoration of continuity. Immediately increasing intestinal surface area for nutrient absorption is possible by recruiting all available bowel by restoring continuity following initial surgery [29].

The procedures may involve closing a diverting ostomy, fixing an enterocutaneous fistula, or reversing bypassed bowel segments. Furthermore, addressing strictures and other obstructions in the bowel can help maximize adaptive potential. Using both small and large bowel offers benefits for both groups. Children who have less than 50% of their expected small bowel length for their age and more than 50% of their large bowel intact have shown a 56% probability of achieving enteral autonomy. The findings of a systematic review in adult population, specifically those with less than 100 cm of small intestine, highlight the importance of maintaining continuity in the colon. This factor was found to significantly contribute to patients achieving full enteral autonomy, emphasizing the critical role of optimizing bowel length regardless of age [30,31].

In cases of rapid intestinal transit without bowel dilatation, surgical alternatives include reversed small bowel interposition grafts or isoperistaltic colonic interposition grafts. The reason behind this is that inserting a section of bowel with peristalsis that is reversed or slower (colon) would increase the amount of time that nutrients and fluids have to be absorbed. These procedures are generally not carried out in children for practical reasons. In younger children who have the potential for gut expansion, the interposed grafts can also elongate and potentially cause intestinal obstruction [29]. In adults, these procedures have shown enhanced intestinal absorption and reduced reliance on parenteral nutrition. However, there have been more frequent reoperations due to the occurrence of surgical problems [7,32,33] (Table 3).

## 8. Intestinal Failure Associated Liver Disease (IFALD)

Although a detailed analysis of IFALD is not within the scope of this review, it is worth mentioning that there is ongoing academic discussion over the diagnostic criteria for this prevalent and potentially fatal complication associated with prolonged PN utilization [34]. Mutanen et al. sought to elucidate the natural progression of IFALD in pediatric patients. A total of 77 children with intestinal failure (IF) who were receiving long-term PN were recruited by their team for liver biopsies. The severity of their IFALD was assessed using histological criteria, which included active IFALD (including inflammation and/or cholestasis), chronic IFALD (involving Metavir fibrosis and steatosis stage ≥ 2), and normal (no IFALD). Out of the original 77 cases, 48% had active disease, 21% exhibited chronic IFALD, and 31% showed no signs of IFALD. Significantly, there was a negative correlation observed between patient age and both cholestasis (r = −0.65, *p* < 0.001) and portal inflammation (r = −0.39, *p* = 0.001). Acute IFALD was linked to notably elevated levels of AST/ALT, GGT, total and direct bilirubin, reduced citrulline, and increased liver stiffness as evaluated by transient elastography. No discernible disparities were seen between chronic IFALD and normal biopsies. A total of forty-eight individuals underwent a further liver biopsy after an approximate duration of 3 years. Out of the 25 individuals who had active IFALD during their initial biopsy and were re-biopsied, 9 (36%) showed no change, 7 (29%) had progressed to chronic IFALD, and 9 (36%) had normal biopsies. Out of the 14 patients who did not have IFALD and had a repeat biopsy, 11 showed no change, two (14%) had active IFALD, and one (7%) had chronic IFALD. Patient with active IFALD exhibited markedly elevated values of AST/ALT, GGT, total and direct bilirubin, along with enlarged spleen size and increased aminotransferase platelet ratio index. The concurrent measurement of GGT and citrulline levels demonstrated a sensitivity of 83% and a specificity of 90% in accurately predicting the presence of active IFALD upon repeat biopsy. This study likely overestimates the prevalence of IFALD advancement since it suffered from selection bias by only getting a second biopsy from individuals who were at the highest risk of liver disease. Utilizing biochemical markers and liver stiffness can provide a valuable, easy, and dependable method for categorizing the risk of patients with chronic PN and predicting the probability of developing and advancing to active IFALD [35].

## 9. Conclusions

Intestinal failure management necessitates a comprehensive, multidisciplinary approach. The primary objective is to enhance intestinal function and patient quality of life while addressing complications such as chronic diarrhea, malabsorption, and IFALD. Advances in medical and surgical therapies, including GLP-2 analogs and optimized parenteral nutrition protocols, have significantly improved outcomes. However, challenges persist, particularly in transitioning pediatric patients to adult care and managing long-term PN-related complications.

The integration of nutritional support with emerging pharmacological treatments and surgical innovations offers promising prospects for further improving patient outcomes. Specialized intestinal rehabilitation centers play a pivotal role in delivering holistic care and improving long-term prognoses for IF patients. Continued emphasis on personalized treatment plans that adapt to the evolving needs of patients throughout their lifespan is essential for optimal management of this complex condition.

## Figures and Tables

**Figure 1 diagnostics-14-02114-f001:**
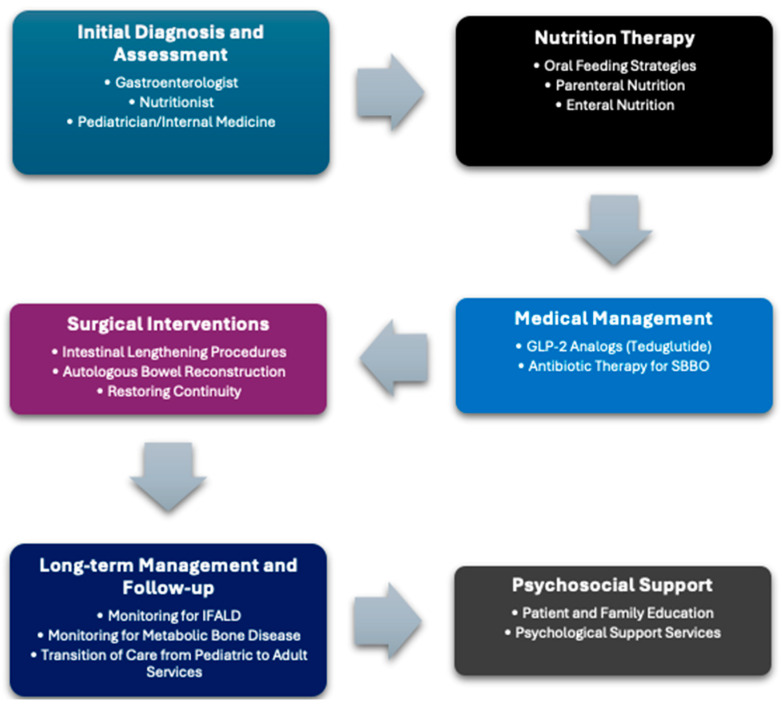
Illustrates the Multidisciplinary Management Approach for Intestinal Failure.

**Table 1 diagnostics-14-02114-t001:** Classification of intestinal failure in adults.

Type	Timing of Presentation	Speed of Onset	Duration
1	Acquired	Acute	<28 days
2	Congenital/acquired	Acute	Weeks to months
3	Congenital/acquired	Chronic	Months to years

**Table 2 diagnostics-14-02114-t002:** Summary of Etiologies and Classifications of Short Bowel Syndrome in Intestinal Failure.

Classification	Description	Primary Etiology	Prevalence	Patient Demographics
Type 1 SBS-J	Short bowel with end jejunostomy	Crohn’s Disease	67%	Predominantly adults, higher in females
Type 2 SBS-JC	Short bowel with jejuno-colic anastomosis	Mesenteric Ischemia	61.4%	Adults aged 50 or older
Type 3 SBS-JIC	Short bowel with jejuno-ileal anastomosis	Crohn’s Disease Mesenteric Ischemia	52.3%	Adults aged 50 or older

**Table 3 diagnostics-14-02114-t003:** Clinical Outcomes and Risk Factors in Pediatric vs. Adult Intestinal Failure.

Outcome/Factor	Pediatric IF	Adult IF
Primary Etiologies	Short Bowel Syndrome (84.9%), Enteropathies (7.9%), Intestinal Dysmotility (7.2%)	Short Bowel Syndrome, Mesenteric Ischemia, Crohn’s Disease
Gender Prevalence	Predominantly male (61.2%)	Predominantly female (68%)
Enteral Autonomy Rate	48%	Varies, lower compared to pediatric
Mortality Rate	Decreasing but still significant	3.8% in-hospital mortality
Risk Factors for Mortality	Catheter-related bloodstream infections, Intestinal Failure Associated Liver Disease, Sepsis, Liver disease, posttransplant status	Sepsis, Age > 65, Acute Malnutrition, Congestive Heart Failure, Simultaneous Liver Disease
Long-term PN Dependence	Higher in pediatric patients transitioning to adulthood	Variable, depending on etiology and management

## Data Availability

Not applicable.

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
