# Peer review of "A Multidisciplinary Approach to the Classification and Management of Intestinal Failure: Knowledge in Progress"

_diagnostics, 2024, doi:10.3390/diagnostics14192114_

Round 1

Reviewer 1 Report

Comments and Suggestions for Authors

This is a comprehensive review on intestinal failure with some new insights. I would suggest restructuring the text by separating definitions for children and adult IF, as well as separating epidemiology for children and adults. I propose adding a separate paragraph on rehabilitation (separate from SIBO), before the paragraphe on surgery. 

Specific points:

1. Epidemiology: : line 110-112: The paragraph on IV support is not very clear. Usually Type 1 requires more IVS and have less rehabilitation and less chances for NP weaning. This part can be better explained and expanded in the rehabilitation paragrpahe

2. Table 3: risk factors for mortality: the elements in the column for paediatric population are mostly protective factors. Please adapt

3. Medical management: I would put this after rehabilitation and medical nutrition. Add some data on the effect of teduglutide and which group responds the best

4. Line 312: Explain enteral fat

5. Oral nutrition with CH in patients with the colon is also important for colon trophicity through short chain fatty acids. Please explain

6. Surgical interventions Figure: add restoring continuity

7. IFALD is important to monitor. Another important one term consequence is Bone health. Please include a para on this aspect

8. Line 343: Patients undergoing surgery with consequent SBS will in a majority need at least temporary PN after leaving the hospital, even if they have adequate caloric intake. Please elaborate

9. Line 352: Maximise oral nutrition, instead of enteral nutrition

Comments on the Quality of English Language

Thorough revision and English editing is required.

Author Response

I would suggest restructuring the text by separating definitions for children and adult IF, as well as separating epidemiology for children and adults.

Response:Following your suggestion, we have better separated the definitions of IF in adults and in the pediatric population, and in the epidemiology section, subsections have been added to better differentiate when talking about each patient population.

I propose adding a separate paragraph on rehabilitation (separate from SIBO), before the paragraphe on surgery. 

Response: A new section has been added (before the surgery section) that briefly discusses rehabilitation in patients with IF.

Specific points:

  1. Epidemiology: : line 110-112: The paragraph on IV support is not very clear. Usually Type 1 requires more IVS and have less rehabilitation and less chances for NP weaning. This part can be better explained and expanded in the rehabilitation paragrpahe

Response: According to Pironi et al. 2021, published in Clinical Nutrition ESPEN on page 437, section 3.2, the following is described: The duration of IVS was shorter in SBS-J (less than 3 years in more than one-half), whereas it was longer than 3 years in 61.4% of SBS-JC and in 52.3% of SBS-JIC. The number of days of IVS infusion per week and the percentage of patients requiring the FE type of IVS were greater in the SBS-J group. The volume of IVS was also higher in SBS-J than in SBS-JC and SBS-JIC, whereas no differences between the SBS types were observed in IVS energy.

We understand that the literature can be controversial, however, we describe and reference one of the most robust and largest studies published in the last 5 years. Therefore, we must consider the limitations of a single study, and take the data with caution.

  1. Table 3: risk factors for mortality: the elements in the column for paediatric population are mostly protective factors. Please adapt

Response: The factors described in the table in relation to the pediatric population have been modified to appropriately mention risk factors for mortality in pediatric patients with IF.

  1. Medical management: I would put this after rehabilitation and medical nutrition. Add some data on the effect of teduglutide and which group responds the best

Response: Following your suggestion, we have changed the structure of the article to follow the order of Figure 1: nutritional therapy, medical management, surgical intervention, and follow-up of complications (IFALD, bone health). In addition, we have added a small paragraph with the approved indication for treatment with teduglutide and the evidence for its efficacy in the pediatric population.

  1. Line 312: Explain enteral fat

Response: We have removed the reference to "enteral fat" because, as you point out, we believe it is unclear.

  1. Oral nutrition with CH in patients with the colon is also important for colon trophicity through short chain fatty acids. Please explain.

Response: In the new rehabilitation section, we have added information regarding the benefits of carbohydrate consumption and its fermentation, in the colon, into short-chain acids for use as an energy source.

  1. Surgical interventions Figure: add restoring continuity

Response: We have added your suggestion to Figure 1.

  1. IFALD is important to monitor. Another important one term consequence is Bone health. Please include a para on this aspect

Response: A subsection has been added to the Parenteral Nutrition section that discusses monitoring bone health and its direct relationship to IF. The need to monitor for metabolic bone disease in patients with IF has also been added to Figure 1.

  1. 8. Line 343: Patients undergoing surgery with consequent SBS will in a majority need at least temporary PN after leaving the hospital, even if they have adequate caloric intake. Pleaseelaborate

Response: Based on your suggestion, we have clarified the above statement.

  1. Line 352: Maximise oral nutrition, instead of enteral nutrition

Response: We have changed the text according to your suggestion.

Reviewer 2 Report

Comments and Suggestions for Authors

It's an interesting review. The following are suggested to enhance the manuscript 

1. Relevant data about various headings should be in a tabular format for better viewing

2. The way forward needs to be highlighted better as the newer insights are not shining through this review 

3. The following may be referenced and discussed:

a. Allan P, Lal S. Intestinal failure: a review. F1000Res. 2018 Jan 18;7:85. doi: 10.12688/f1000research.12493.1. PMID: 29399329; PMCID: PMC5773925.

Comments on the Quality of English Language

The English language is largely acceptable 

Author Response

  1. Relevant data about various headings should be in a tabular format for better viewing

Response: Thank you in advance for your recommendation. Adjustments were made to the layout, wording and presentation of information throughout the article.

  1. The way forward needs to be highlighted better as the newer insights are not shining through this review 

Response: Thank you in advance for your recommendation. Therefore, we submit for your consideration, the possible change of the title to align the objective of the text with the current state of the evidence. The proposed title is: A Multidisciplinary Approach to the Classification and Management of Intestinal Failure: Knowledge in Progress

  1. The following may be referenced and discussed: Allan P, Lal S. Intestinal failure: a review. F1000Res. 2018 Jan 18;7:85. doi: 10.12688/f1000research.12493.1. PMID: 29399329; PMCID: PMC5773925.

Response: We appreciate your suggestion. We have added the reference you mentioned, which was very helpful for some of the newly added paragraphs (in the new Rehabilitation section, as well as where we discuss metabolic bone disease, in the Parenteral Nutrition section).